# Influence of Non-Occupational Physical Activity on Burnout Syndrome, Job Satisfaction, Stress and Recovery in Fitness Professionals

**DOI:** 10.3390/ijerph18189489

**Published:** 2021-09-08

**Authors:** María Jesús Marín-Farrona, Manuel León-Jiménez, Jorge García-Unanue, Leonor Gallardo, Gary Liguori, Jorge López-Fernández

**Affiliations:** 1IGOID Research Group, Department of Physical Activity and Sport Sciences, University of Castilla-La Mancha, 45071 Toledo, Spain; mmarinfa@gmail.com (M.J.M.-F.); manuel.leonjimenez@yahoo.es (M.L.-J.); Leonor.Gallardo@uclm.es (L.G.); 2College of Health Sciences, University of Rhode Island, 55 Lower College Road, Kingston, RI 02881, USA; gliguori@uri.edu; 3Centre for Sport, Exercise and Life Sciences (CSELS), Coventry University, Coventry CV1 5FB, UK; lopezfej@uni.coventry.ac.uk or; 4School of Sport Sciences, Universidad Europea de Madrid, 28670 Madrid, Spain

**Keywords:** heart rate variability, personal trainer, CESQT questionnaire, HRV biofeedback, mobile health

## Abstract

Background: This study aimed (1) to analyse the effect of non-occupational physical activity (NOPA) on the stress levels of fitness professionals, and (2) to apply a questionnaire to workers measuring burnout syndrome, working conditions and job satisfaction, and to compare the results with physiological stress and recovery measured objectively through heart rate variability (HRV). Methods: The HRV of 26 fitness instructors was recorded during 2–5 workdays using Firstbeat Bodyguard 2. Participants also completed a questionnaire (CESQT) measuring working conditions and job satisfaction variables and occupational burnout syndrome. Results: NOPA showed a negative association with both the percentage of stress (*p* < 0.05) and stress–recovery ratio (*p* < 0.01), and a positive association with the percentage of recovery (*p* < 0.05). Better work conditions (working hours, salary satisfaction and length of service) were associated with lower stress in fitness professionals. Conclusion: NOPA appears to improve the stress levels of fitness instructors in this study cohort. Self-reported burnout levels measured through the CESQT questionnaire do not coincide with the physiological stress responses measured through HRV. Better working conditions appear to reduce the stress response in fitness professionals.

## 1. Introduction

Stress, defined as the physical and mental responses of the body and the adaptations to perceived changes in life [1], is a potential risk factor on wellbeing and mortality [2,3]. Work stress, also known as “burnout syndrome,” was first classified as mental ill-health at the workplace in the 11th revision of the International Classification of Diseases [4] and currently affects around 26% of workers [5]

Excessive levels of stress might affect the human nervous system, and thereby the process of memory, cognition and learning [6,7]. At the workplace, work-related stress may result in decreased employee wellbeing, lower worker productivity, or increased rates of absenteeism [2]. In fact, the economic cost of burnout syndrome within European countries has been estimated to be €627 billion [8]. Accordingly, the interest in measuring employees’ stress levels at the workplace has increased in recent years [9].

There are numerous questionnaires available that measure different parameters related to stress at work. However, these questionnaires typically focus on subjective parameters and not on physiological responses [10]. These questionnaire-based limitations can be resolved by recording physiological responses to stress [11], best done through heart rate variability (HRV). HRV is the fluctuation of the length of heartbeat intervals over time (RR-intervals) and has proved to be a valid indicator of physiological stress in different contexts such as clinical patients or the workplace [12,13]. On the one hand, when humans are faced with a stressful situation, the sympathetic nervous system is activated, resulting in an increased HR and shorter RR-intervals. Therefore, low HRV levels are a marker of high stress. On the other hand, while humans feel relaxed, the vagal response is activated as well as the parasympathetic nervous system, resulting in a decreased HR and longer RR-intervals. Thus, high HRV levels are a marker of less stress. Accordingly, HRV represents the ability of the heart to respond to different contextual stimuli [14,15]. Nonetheless, occupational stimuli such as high levels of noise or vibrations might alter the sympathetic and parasympathetic responses in airport staff or students [16,17]. HRV can be assessed by the standard deviation of normal R–R intervals while vagal activity can be determined by the root mean square of successive normal R–R interval differences (RMSSD) using a non-intrusive and pain-free instrument. Thus, the higher the HRV or more activation of the parasympathetic nervous system, the greater the RMSSD values [18].

The identification of stress levels at the workplace and the application of HRV biofeedback exercises (i.e., breathing with a HRV biofeedback device) seem to be an effective way to reduce stress-related diseases and promote health [13,19]. Furthermore, HRV has been used to monitor and mitigate burnout syndrome in different populations such as white-collar workers, police officers, nurses, managers and health professionals [20]. Thus, monitoring HRV can be an important part of designing strategies for health assessment and health promotion, particularly regarding reducing levels of stress and anxiety.

Besides biofeedback exercises, physical activity (PA) conducted either at the worksite or in leisure time might be effective in reducing stress levels [21]. This is due to the autonomic nervous system (ANS), which controls cardiovascular function through sympathetic and parasympathetic modulation and is active during and after PA to maintain homeostasis. Repeated exposure to PA elicits physiological adaptations that reduce homeostatic perturbation in response to further stressors and therefore, increase HRV [22,23]. Consequently, regular PA is considered an effective tool to reduce perceived burnout and non-occupational stress in the general population [24]. Furthermore, attempts to reduce employee stress and burnout syndrome through workplace physical activity (PA) programs have reported promising results [25,26].

Fitness instructors are a special population to study because their job includes performing several hours of moderate-to-vigorous PA (MVPA) per day [27], while most adults only engage in a few minutes of MVPA daily [26]. Thus, occupational PA performed by fitness instructors may not result in a decrease in stress, anxiety and work-related burnout as occurs in other professions [27], and further PA (i.e., non-occupational PA (NOPA)) might be needed to produce these benefits [28]. However, to the best of the authors’ knowledge, no-one has studied the relationship of NOPA with daily stress levels measured through HRV and burnout perception in this population. In fact, the few existing studies in fitness instructors only describe stress-related variables such as job satisfaction and working conditions, so further research in this population is needed [29]. Thus, the objectives of this study were (1) to analyse the effect of NOPA on the stress levels of fitness professionals, and (2) to apply a questionnaire to workers measuring burnout syndrome, working conditions and job satisfaction, and to compare these results with physiological stress and recovery measured objectively through heart rate variability (HRV). We hypothesized that NOPA would decrease the stress levels of fitness professionals.

## 2. Materials and Methods

### 2.1. Participants

The final sample was composed of 26 fitness instructors (18 men and 8 women; 33.08 ± 8.15 years) from a total of 38 instructors recruited, or a 68% response rate. The workers were part of the staff of two separate fitness centres, nine from one centre and 17 from the other. Participants agreed to voluntarily participate in this study and reported never having been diagnosed with heart disease, high blood pressure, diabetes or any other chronic disease [19]. The fitness instructors were required to give at least 3 to 4 h/day of supervised instruction and record their heart rate for 48 h.

All the participants were informed of the risks related to the study, the data protection procedure and the objective of the study. All participants signed the informed consent form before participating in the study. This study complies with the ethics committee of the Health Sciences Research Committee of the European University (CIPI/045/16), based on the Helsinki declaration.

### 2.2. Design

This is a transversal and correlational study. The design of this study is displayed in Figure 1. All participants had (1) to keep a self-reported diary where they indicated all the activities performed during the registration time (working hours and sleeping time), (2) to partake in 3–4 h/day of supervised classes, (3) to complete a wellbeing questionnaire which included working conditions and job satisfaction variables (first part) and the CESQT questionnaire (second part), and (4) to monitor their HRV through a portable device. Those who did not meet these requirements, or who did not record their heart rate for 48 h, or who reported an error >15% in HR measurement, or who lost 30 min of recording within 24 h, were excluded from the analysis.

### 2.3. Procedure

#### 2.3.1. Body Mass Index (BMI)

Weight and height were registered using a scale/height SECA scale measurement (model 711; SECA GmbH & Co, KG, Hamburg, Germany). Body mass index (BMI) was calculated using the formula (weight(kg)/height(m)^2^).

#### 2.3.2. Heart Rate Variability (HRV)

The HRV measurements were recorded using the Firstbeat Bodyguard 2 device (Firstbeat Technologies Ltd., Jyväskylä, Finland), a non-invasive device developed by Firstbeat Technologies Ltd. This instrument has been used in previous studies [1,14] and has shown acceptable validity and reliability for the variables included in this study [30].

The device registers the HRV in beats per minute, obtained through the measurement of the fluctuation of the duration of heartbeat intervals (R–R intervals). From this value, the root mean square of standard deviation (RMSSD (in milliseconds)), RMSSD 4 h (in milliseconds), stress percentage, recovery percentage, stress–recovery ratio and stress balance variables were obtained. (Table 1). These are the variables provided by the software associated with the device Bodyguard 2. RMSSD 4 h was measured in the first 4 h of sleeping time, excluding the first half-hour in order to avoid disturbances in the measurement, as slow-wave sleep usually takes place during this time [1]. In addition, the participants agreed to record their hours of work and sleep in a diary in order to compare them with the total recorded data.

#### 2.3.3. Wellbeing Questionnaire

A two part ad hoc auto-administrated online questionnaire was given to the fitness instructors. The first part was formed by three dichotomous and categorical questions related to job satisfaction and working conditions (Table 2) that have been used as indicators to measure job satisfaction in other research with the same sample [31]. The second part was formed by the CESQT questionnaire (Questionnaire for Evaluating Burnout Syndrome) which has been validated in previous studies [31]. The questionnaire is made up of 20 items that are scored using a Likert scale with frequency response format of 5 points: from never (0) to every day (5). The items of the questionnaire are distributed in the following dimensions: work enthusiasm, psychological burnout, indolence and guilt. A total score (CESQ score) is obtained using the following formula: (20- Enthusiasm + Burnout + Indolence)/15 [32].

### 2.4. Statistical Analysis

The results are presented as average and standard deviations. The normality of the variables was tested using the Kolmogorov–Smirnov (KS) test. The relationship between job satisfaction, working conditions, subjective stress and HRV variables was assessed by a series of multivariate linear regression models. A model was estimated for each dependent variable, represented by six indicators of physiological stress (percentage of stress, percentage of recovery, stress–recovery relationship, stress balance, RMSSD and RMSSD 4 h). The factors related to the work and subjective stress of the participants were used as independent variables. All the models were subjected to the variance inflation factor (VIF) test, demonstrating the nonexistence of collinearity, as well as the test for heterozadastity and normality of the residuals, complying with the necessary assumptions for the estimates. The sports centre and gender were included in the model as covariables in the form of a dummy variable. These analyses were conducted using STATA version 14.0. The significance level was established at *p* < 0.05.

## 3. Results

Table 3 displays the descriptive statistics from the variables used in the study. The average stress percentage and recovery percentages are within the optimal values according to the patterns provided by the manufacturer of the HRV monitors (Firstbeat Technologies Ltd., Jyväskylä, Finland).

Although the reliability of the burnout questionnaire has already been checked in previous studies, a reliability analysis was done using alpha Cronbach with the sample of this study for each one of the dimensions which obtained values higher than 0.7 (enthusiasm: α = 0.910; burnout: α = 0.765; indolence: α= 0.796; guilt: α = 0.796).

The multiple regression analysis is shown in Table 4. The centre presented significant values on the stress percentage model (*p* < 0.05), stress–recovery ratio (*p* < 0.05) and stress balance (*p* < 0.01). The gender showed a significative association with stress balance (*p* < 0.05), with men obtaining lower results in this variable. A negative association between age and heart rate variability was found. An increase of 1 year in age leads to an average increase of 2.36 in the RMSSD in sleep (*p* < 0.01) and 1.96 in rest during the first 4 h after sleep conciliation (*p* < 0.05). BMI showed a negative influence on stress balance (*p* < 0.05) but positive on RMSSD (*p* < 0.05).

Regarding the first part of the wellbeing questionnaire, the length of the service is positive when related to the stress percentage (*p* < 0.05) and the stress–recovery ratio (*p* < 0.05), but negative when related to stress balance (*p* < 0.05). More weekly working hours (taking as a reference the average; 36.88 h) is negatively associated with the stress–recovery ratio variable (*p* < 0.05) but positively associated with both RMSSD variables (*p* < 0.01). A higher remuneration satisfaction has a positive influence on stress balance (*p* < 0.01) and on the RMMSD (*p* < 0.05).

The hours of leisure-time PA (considering that all participants performed at least 3 to 4 h of occupational PA), influence the stress percentage (*p* < 0.05) and stress–recovery ratio (*p* < 0.01) negatively, but the recovery percentage (*p* < 0.05) positively. Finally, the CESQT questionnaire score shows a significant and negative influence on the stress percentage (*p* < 0.05) and the stress–recovery ratio (*p* < 0.05), as well as a positive influence on the recovery percentage (*p* < 0.01) and the stress balance (*p* < 0.05).

## 4. Discussion

This is the first study analysing the effect of NOPA and wellbeing variables on the physiological stress levels of fitness instructors. The main results suggest that NOPA, length of service, working hours and salary satisfaction impact stress levels favourably in fitness professionals. Moreover, the CESQT questionnaire did not replace the measurement of physiological stress.

According to our first objective, the results from the regression analysis suggest that greater levels of NOPA led to lower stress levels and higher recovery levels. These results are in line with previous studies that reported a positive influence of NOPA on occupational stress management [24]. Other authors demonstrated that additional PA, above the recommended 150 min of moderate-to-vigorous PA/week, was associated with a higher HRV during workdays and during working hours, which means that an individual finds it easier to suffer less stress [33]. Therefore, despite the amount of PA performed by fitness instructors at the workplace, NOPA seems to provide additional help in stress management.

The scientific literature describes how sympathetic and parasympathetic nervous system response is reflected in HRV. However, a recent study of the effect of different sources of occupational stress on HRV showed no clear sympathetic and parasympathetic system response to some occupational stimulus. For example, whole-body vibration, which in some cases has been used as a training method, produces parasympathetic as well as sympathetic nervous system activation. Thus, care has been taken when considering our results, as it is difficult to draw conclusions regarding which occupational stimuli actually increase stress levels in our studied population [16].

A higher BMI level was associated with a worse RMSSD and therefore higher levels of stress. This value coincides with other authors [34], who found that higher BMI in young adults negatively impacts overall HRV and parasympathetic activity. However, further research is needed because fitness professionals usually have a higher percentage of muscle mass than the general population due to, among other things, their professional activities [35]. Therefore, higher values of BMI in this population are mostly due to greater muscle mass instead of fat mass, as would be typical in the general population. A higher BMI has been associated with a lower amount of recovery during sleep, but in population with high levels of fat [19]. In the present research, this relation was not significant, probably because in this population the high BMI values are due to high muscle mass levels. Therefore, further research assessing variables such as the percentage of fat and muscle mass would provide further information on the relationship between body composition and stress.

Concerning our second objective, the CESQT, which measures burnout syndrome, showed a negative association with HRV, so the hypothesis is rejected. Accordingly, the CESQT questionnaire may not be sensitive enough to assess the stress of a fitness professional, as there are other variables associated with job satisfaction that could impact the results. Therefore, the CESQT questionnaire does not replace the measurement of physiological stress. This result is in accordance with other studies [36] that concluded that although there is a growing body of literature on the physiological correlates of clinical burnout, there are no biomarkers to date for the measurement of this condition. Consequently, it is recommended that both tools, the CESTQ questionnaire and HRV measurements, be integrated to detect stress levels. With respect to the variables related to working conditions and job satisfaction, the length of service was found to have a stronger positive relationship with physiological stress levels. This is in line with a previous research that analysed the influence of job satisfaction on the stress levels of fitness professionals [37]. The number of working hours during the week was analysed and suggests that more weekly working hours are positively associated with both RMSSD variables, meaning that working more hours during the week can decrease the stress levels of fitness professionals. According to other contributors [38], workforce turnover in the fitness professions is high as a result of low pay and the prevalence of shift work, among other factors. Therefore, promoting a full working day without shift work could improve the levels of stress and job satisfaction of fitness professionals. The salary was also studied, and the results suggest that higher remuneration satisfaction has a positive influence on RMMSD. Based on these findings, it can be concluded that improving workplace conditions might have a positive effect on workers’ levels of health, happiness, subjective wellbeing and self-esteem [36].

Interestingly, HRV can be measured through different tools (i.e., electrocardiogram and plethysmography). Among them, plethysmography using a smartphone is a low-cost method to get HRV-related biofeedback in humans [21]. Although it is less accurate than an electrocardiogram, it can overcome the limitation of self-reported data from questionnaires and target a large population [39]. Accordingly, future interventions in workers might consider measuring the efficacy of NOPA on workers’ stress using smartphones to support workers in gaining voluntary control over real-time based physiological processes involved in stress [39]. This could address one of the main limitations of this research, since being a less invasive tool than the electrocardiogram, it may be accessible to more people.

Finally, the authors acknowledge that the present study is limited by its relatively small number of participants. A larger sample will be necessary to make generalizations. Furthermore, PA has been measured subjectively, while exercise intensity was not assessed at all. Thus, further research should try to measure both variables in an objective way. In addition, further analyses on the effect of occupational stressors on fitness instructors’ HRV should be conducted to learn the specific responses of the sympathetic and parasympathetic systems to these stimuli.

## 5. Conclusions

This research suggest that NOPA is an effective tool for managing work-related stress in fitness professionals, but its findings cannot be generalised to other population groups that also conduct high levels of PA at work. (e.g., riders, farmers, firefighter). Self-reported stress does not coincide with the physiological stress responses measured by HRV, thereby suggesting that they measure different dimensions. Finally, better working conditions appear to reduce the stress response in fitness professionals.

## Figures and Tables

**Figure 1 ijerph-18-09489-f001:**
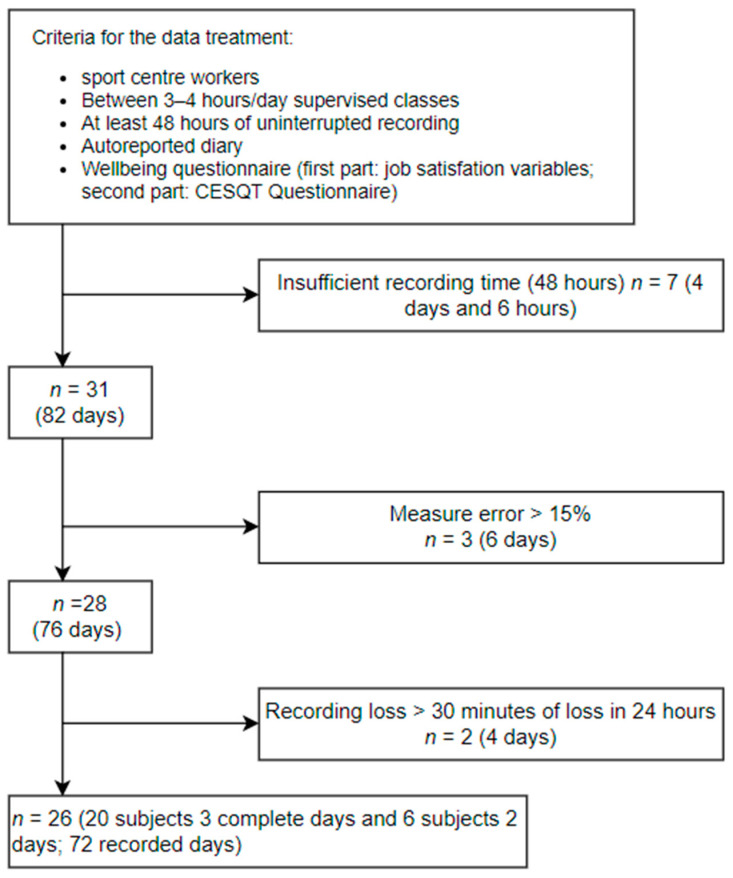
Flowchat describing the study design.

**Table 1 ijerph-18-09489-t001:** Description of the variables.

Variable	Definition
**Heart rate variability**	
RMSSD (ms)	Average every 5 min of the RMSSD which is the square root of the mean of the union of the adjacent R–R intervals. Measured during sleep. This is a vagal heart control indicator (parasympathetic tone). The higher the RMSSD, the lower the physiological stress and higher health.
RMSSD 4h	Average minute by minute of the RMSSD during the first 4 h of sleep after the first half an hour of sleep.
**Stress and recovery**	
% Stress	Indicates the percentage of time between the stress reactions within/for 24 h.
% Recovery	Indicates the percentage of time between the recovery reactions within/for 24 h.
Ratio stress-recovery	Ratio of the division between % of stress and % of recovery, where higher values mean higher stress levels in 24 h.
Stress Balance	The difference between the total time classified as recovery during sleep and the total time classified as stress during sleep, divided by the sum of total time classified as recovery during sleep and of the total time classified as stress during sleep. Values from 0.5 to 1 indicate a good recovery; values of 0 to 0.5 indicate moderate recovery and values of 0 to −1 indicate insufficient recovery.

RMSSD = root mean square of the successive difference.

**Table 2 ijerph-18-09489-t002:** First part of the questionnaire: dichotomous and categorical questions.

Variable	Definition
Length of service (months)	Time in the job position.
Working hours (hours)	Hours of work a week.
Salary Satisfaction	Global evaluation of the worker in relation to their economic satisfaction with their salary. Scale of 4 points.
Non-occupational PA (hours)	Number of hours a week practising PA out of the working hours.

PA = Physical activity.

**Table 3 ijerph-18-09489-t003:** Outcomes form the studied variables.

Variables	Mean ± SD		
**BMI**	24.67	±	3.29
**Stress and Recovery**			
% Stress	50.23	±	9.79
% Recovery	32.79	±	8.68
Ratio stress- recovery	1.78	±	0.81
Stress Balance	0.79	±	0.14
**Heart Rate Variability**			
RMSSD	60.40	±	29.64
RMSSD4h	59.35	±	32.46
**Wellbeing questionnaire**			
First part—Job satisfaction variables			
Length of service	49.23	±	52.07
Working hours	36.88	±	9.42
Salary satisfaction	2.19	±	0.94
Non-occupational PA (hours)	5.77	±	2.69
Second part—CESQT questionnaire			
CESQT questionnaire	15.88	±	7.81
Enthusiasm	16.77	±	4.95
Burnout	6.31	±	2.88
Indolence	6.65	±	4.40
Guilt	4.42	±	3.59

Data are presented as a mean ± SD. Abbreviation: BMI = Body mass index; CESQT = Questionnaire for the evaluation of occupational burnout syndrome; PA = Physical activity; RMSSD = Root mean of successive standard deviation.

**Table 4 ijerph-18-09489-t004:** Multiple regression analysis of the centre variable, physical variables, working conditions and subjective stress on objective stress variables (standard errors in brackets).

Variables	% Stress		% Recovery		Ratio s/r		Stress Balance		RMSSD		RMSSD 4h	
centre	8.874	(3.969) *	−5.429	(4.077)	0.784	(0.335) *	−0.153	(0.047) **	−10.298	(9.053)	−10.746	(12.070)
sex	−9.242	(4.387)	2.812	(4.506)	−0.254	(0.371)	−0.122	(0.052) *	2.261	(10.006)	−1.089	(13.341)
age	−0.019	(0.267)	0.197	(0.274)	−0.017	(0.023)	0.002	(0.003)	−2.362	(0.610) **	−1.958	(0.813) *
non-occupational pa	−2.001	(0.771) *	1.808	(0.792) *	−0.194	(0.065) **	0.018	(0.009)	−0.589	(1.759)	−1.296	(2.346)
bmi	0.785	(0.651)	−0.975	(0.669)	0.071	(0.055)	−0.019	(0.008) *	3.267	(1.485) *	3.237	(1.980)
length of service	0.116	(0.041) *	−0.063	(0.042)	0.008	(0.003) *	−0.001	(0.000) *	−0.086	(0.094)	−0.137	(0.126)
working hours	−0.404	(0.212)	0.382	(0.218)	−0.045	(0.018) *	0.004	(0.003)	1.504	(0.483) **	1.686	(0.645) *
salary satisfaction	−1.874	(1.784)	1.584	(1.833)	−0.185	(0.151)	0.069	(0.021) **	8.686	(4.069) *	11.213	(5.426)
cesqtquestionnaire	−0.585	(0.249) *	0.726	(0.255) *	−0.066	(0.021) **	0.009	(0.003) *	−0.853	(0.567)	−0.315	(0.756)
constant	56.462	(20.200) *	20.321	(20.750)	3.436	(1.708)	0.963	(0.238) **	17.759	(46.076)	−8.487	(61.434)
r^2^ *	0.580		0.437		0.547		0.636		0.560		0.647	

Data are presented as a mean and * *p* < 0.05; ** *p* < 0.01. Abbreviations: Ratio S/R = Ratio Stress–Recovery.

## Data Availability

The data presented in this study are available on request from the corresponding author. The data are not publicity available due to participants did not accept their data to be share with third bodies or people besides the research team.

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
