# Peer review of "Influence of Non-Occupational Physical Activity on Burnout Syndrome, Job Satisfaction, Stress and Recovery in Fitness Professionals"

_ijerph, 2021, doi:10.3390/ijerph18189489_

Round 1

Reviewer 1 Report

The paper ist well witten with some shortcomings.

Major comments:

The study has only a very small N.

Thus some of the conclusions and also some of the analysis seem to be too ambitious.

Conclusions/Discussion: there are onyl 26 subjects, all being fitness professionals, thus conclusions (or generalisations in discussion) should be limited only to that group (not for employees or working conditions in general).

Analysis: having only 26 subjects, the regressions models using 9 predictor candidates seem to be very/too ambitious. There is a risk of biased estimators. I would like to see the bivariate relations of the 9 predictors to the 6 outcomes in table 5. Probably some bivariate relations are reversed in the multivariate models.

And it would be helpful to state, in which outcomes high values are positive and in which low values - that makes it easier to interprete regression coeffs..

To few information is given on the questionnaire on job satisfaction. What are the questions? The title "job satisfaction" does not seem correct, as the questions are only partly on that topic. Where does the questionnaire come from? Why did the authors not use a validated tool for job satisfaction and working conditions and percievew stress (like COPSOQ or ERI oder JDR)?

Minor comments:

Error in line 71 reference

Check formula in line 112. I think it should be: (weight(kg) / height(m)²)

Check formula in line 137. It seems uncomplete.

line 149 is it two-tailed?

line 155-160 seems double

In statistical analysis "outcome" means dependent varibale. Thus this word should not be used to decribe scales or constructs (see line 156, 192)

line 162-166: center is a dummy of the two centres. Thus, the direction of the regression coeff. has no meaning ("positive effect"). Coding the 2 centres the other way would lead to a negative effect. And, using centre as a covariate does not improve the robustness of the model, I do not understand why this should be an effect?!

good luck

Author Response

Dear reviewer,

Thank you for all your comments and indications for improvements. Undoubtedly, these are contributions of high quality, which have made the article more valuable.

We have tried our best to answer each question raised (in bold). In addition, the attached document shows the changes and modifications (in red).

The paper ist well witten with some shortcomings.

Major comments:

The study has only a very small N.

Thus some of the conclusions and also some of the analysis seem to be too ambitious. Thank you very much for this comment, we have changed the language in which we present the results obtained. We have made a greater number of suggestions instead of statements and generalizations.

Conclusions/Discussion: there are only 26 subjects, all being fitness professionals, thus conclusions (or generalisations in discussion) should be limited only to that group (not for employees or working conditions in general). Thank you very much for your comment, discussion and conclusion has been rewritten accordingly to avoid mentions other collectives and generalizations. For example, with the following state: “The main results suggest that NOPA, the length of service, working hours, and salary satisfaction impact favourably on stress levels in fitness professional” (line 204-206). However, future research on other groups and with larger samples should be conducted. “This research suggest that NOPA is an effective tool to manage work-related stress in fitness professionals, but findings cannot be widespread to other population groups that also conduct high levels of PA at work. (e.g., riders, farmers, firefighter)” (line 282-285).

Too few information is given on the questionnaire on job satisfaction. What are the questions? The title "job satisfaction" does not seem correct, as the questions are only partly on that topic. Where does the questionnaire come from? Why did the authors not use a validated tool for job satisfaction and working conditions and perceive stress (like COPSOQ or ERI odder JDR)? Thanks, we appreciate the comment. Perceived stress was measured with the Questionnaire for the Evaluation Burnout Syndrome (CESQT), which has been validated (Gil-Monte, 2014).

Gil-Monte, P.R.; García-Juesas, J.A.; Núñez, E.; Carretero, N.; Roldán, M.D.; Caro, M. Factorial validity of the “Questionnaire for the Evaluation of Burnout Syndrome” (CESQT). Psiquiatria. 2014, 3(6), 7

Regarding job satisfaction, we decided to group these questions into this variable because working hours, salary and length of service have been used as indicators to measure job satisfaction in other research with the same population (line 144-146). Also, we have specified that this is an ad-hoc questionnaire. In addition, due to the characteristic of the job, it was difficult to make another questions (for example, money earned).

Ramos, L. R., Esteves, D., Vieira, I., Franco, S., & Simões, V. (2021). Job Satisfaction of Fitness Professionals in Portugal: A Comparative Study of Gender, Age, Professional Experience, Professional Title, and Educational Qualifications. Frontiers in psychology, 11, 621526. https://doi.org/10.3389/fpsyg.2020.6215267

Minor comments:

Error in line 71 reference: Thanks, it has been addressed 

Check formula in line 112. I think it should be (weight(kg) / height(m)²): Thanks, it has been corrected

Check formula in line 137. It seems uncomplete. We appreciate the comment to the reviewer, but the formula appears is the same that the official document of the questionnaire (Gil Monte et al., 2014)

Gil-Monte, P.R.; García-Juesas, J.A.; Núñez, E.; Carretero, N.; Roldán, M.D.; Caro, M. Factorial validity of the “Questionnaire for the Evaluation of Burnout Syndrome” (CESQT ). Psiquiatria. 2014, 3(6), 7

line 149 is it two-tailed? Yes

line 155-160 seems double: Thanks, this paragraph has been rewritten  

In statistical analysis "outcome" means dependent variable. Thus this word should not be used to describe scales or constructs (see line 156, 192). Thank you. We have replaced the word in these lines.

line 162-166: center is a dummy of the two centres. Thus, the direction of the regression coeff. has no meaning ("positive effect"). Coding the 2 centres the other way would lead to a negative effect. And, using centre as a covariate does not improve the robustness of the model, I do not understand why this should be an effect?! We appreciate the comment to the reviewer. We have written the output paragraphs related to that variable, so as not to cause confusion.

Analysis: having only 26 subjects, the regressions models using 9 predictor candidates seem to be very/too ambitious. There is a risk of biased estimators. I would like to see the bivariate relations of the 9 predictors to the 6 outcomes in table 5. Probably some bivariate relations are reversed in the multivariate models. And it would be helpful to state, in which outcomes high values are positive and in which low values - that makes it easier to interpreted regression coeffs. The estimated regressions are sufficiently robust, being able to add additional information. The Variance Inflation Factor (VIF) remained below 2 in all the variables for all the models (10 is considered as a critical number), showing the absence of multicollinearity. Similarly, the hottest showed the absence of heteroscedasticity problems in all of the models.

Finally, the residues showed normal behaviour. Therefore, the regression results are efficient despite having a small sample. In that case, it is better to use a regression to observe the relationships between variables in the presence of other explanatory factors. The authors have included mention of these assumptions in the methodology section.

Reviewer 2 Report

In this article, Authors have investigated change of HRV parameters related to physical activity, occupational stress and burnout in a sample of professional trainers. The results showed that physical activity improves the level of stress, while burnout does not affect HRV response.  

Limitations: theoretical and practical implications are questionable and poorly discussed.

Strengths: The authors have deepened a current and interesting theme and there is a need to study this issue. The sample size is adequate.

The theme is topical, but several concerns have to be solved before re-evaluating the manuscript and possibly considering it for publication.

Abstract: the results are not clearly presented. State what happens to HRV parameters, instead of a generic “stress response”. Which HRV values where associated and how (increasing / decreasing)?

Introduction need some improvements. The aim of your research should be highlighted more in detail. Also, study motivation and a clear literature gap are not properly reported.  

You stated: “These changes result in a decreased HRV during the presence of the stressor”. This statement is only partially correct. In fact, some of HRV parameters decrease as a response of stressors, while some other parameters (measuring sympathetic activity) can increase due to stress. Be clear reporting how HRV parameters change as a response of occupational and environmental stress sources. About this aspect, Authors can also refer to these articles:

Jalilian H, Zamanian Z, Gorjizadeh O, Riaei S, Monazzam MR, Abdoli-Eramaki M. Autonomic nervous system responses to whole-body vibration and mental workload: A pilot study. Int J Occup Environ Med 2019. https://doi.org/10.15171/ijoem.2019.1688.

Lecca LI, Marcias G, Uras M, Meloni F, Mucci N, Larese Filon F, Massacci G, Buonanno G, Cocco P, Campagna M. Response of the Cardiac Autonomic Control to Exposure to Nanoparticles and Noise: A Cross-Sectional Study of Airport Ground Staff. Int J Environ Res Public Health. 2021 Mar 3;18(5):2507. doi: 10.3390/ijerph18052507. PMID: 33802520; PMCID: PMC7967637.

Hjortskov N, Rissén D, Blangsted AK, Fallentin N, Lundberg U, Søgaard K. The effect of mental stress on heart rate variability and blood pressure during computer work. Eur J Appl Physiol 2004. https://doi.org/10.1007/s00421-004-1055-z.

de la Vega R, Jiménez-Castuera R, Leyton-Román M. Impact of Weekly Physical Activity on Stress Response: An Experimental Study. Front Psychol 2021;11:608217. https://doi.org/10.3389/fpsyg.2020.608217.

You stated: “Using HRV to monitor and 60 mitigate burnout syndrome in different populations […]” It is not clear how HRV monitoring can decrease levels of stress. Explain what is the mechanism involved and working scenarios where HRV monitoring has improved workers well-being reducing stress. How can a monitoring improve well-being? This assertion is very questionable.

Line 71: insert citation

The link between PA, stress levels and HRV changes is not explain at all. The theoretical background that drive your study is very weak and badly presented. Also, the hypothesis are too general and not linked with a strong theoretical framework. I suggest to carefully revise the introduction, providing the needed elements to justify your study. The effect of PA on stress and HRV must be properly presented. It is well known that PA affect autonomic response per se. So, How can attribute changes of HRV to stress and not to PA?

Objective 3 is not properly described. What do you want to evaluate and how?

Methods:

How did you select the variables of interest? Explain very clearly how did you choose HRV parameters.

I noticed that only RMSSD is a validated HRV measure. The other variables such as % stress, % recovery and so on are not validated HRV variables. You must provide the validation of those variables, if any.

Why did you include sport centres as covariates? Do you hypothesised that HRV changes on function of working centre?

Results

The reference you used to compare stress levels whit standard is not a scientific reference (“Firstbeat Technologies Ltd., 153 Jyväskylä, Finland”). You must provide a scientific validated reference, if any, and not a manual of use of the instrument. Is there any standard for this measure? I strongly doubt on the validity of these measures.

Results

I suggest to revise the results of the multiple linear regression. First of all, all the p values reported in the text as <0.05 are not the same of those reported in table 5.

Table 5: you must report standardized beta and p values. The mean values are not required.

Discussion

Overall, the results are poorly discussed. I suggest to follow  STROBE guidelines for cross sectional studies, with those elements:

  • Summarize key results with reference to study objectives
  • Discuss limitations of the study, taking into account sources of potential bias or imprecision.
  • Give a cautious overall interpretation of results considering objectives, limitations, multiplicity of analyses, results from similar studies, and other relevant evidence.
  • Discuss the generalizability (external validity) of the study results

I suggest to revise the interpretation of RMSSD changes. RMSSD mainly reflect parasympathetic activity, so an increase of RMSSD does not reflect an increase of stress. (see also Task Force of the European Society of Cardiology and the North American Society of Pacing and Electrophysiology. Heart Rate Variability : Standards of Measurement, Physiological Interpretation, and Clinical Use. Circulation 1996. https://doi.org/10.1161/01.CIR.93.5.1043.

Shaffer F, Ginsberg JP. An Overview of Heart Rate Variability Metrics and Norms. Front Public Heal 2017;5:258. https://doi.org/10.3389/fpubh.2017.00258.)

Overall, the results are not fully discussed, with a proper comparison with the field literature. A discussion of the generalizability should be highly appreciated in this section. You should also carefully explain what is the specific contribution that your findings bring to literature and knowledge in this area. Please be very clear on what your study adds, exactly how it extends previous knowledge.

Conclusions are only partially supported by your results. In details, HRV is affected by a lot of occupational and environmental factors, so a weak correlation between self-reported stress and HRV is not surprising. Increasing work demand to manage stress is against a lot of previous knowledge that clearly assert that increasing job demand strongly affect perceived stress.

Author Response

Dear reviewer,

Thank you for all your comments and indications for improvements. Undoubtedly, these are contributions of high quality, which have made the article more valuable.

We have tried to answer each question raised (in bold). In addition, the attached document shows the changes and modifications (in red).

In this article, Authors have investigated change of HRV parameters related to physical activity, occupational stress and burnout in a sample of professional trainers. The results showed that physical activity improves the level of stress, while burnout does not affect HRV response.  

Limitations: theoretical and practical implications are questionable and poorly discussed.

Strengths: The authors have deepened a current and interesting theme and there is a need to study this issue. The sample size is adequate.

The theme is topical, but several concerns have to be solved before re-evaluating the manuscript and possibly considering it for publication.

Abstract: the results are not clearly presented. State what happens to HRV parameters, instead of a generic “stress response”. Which HRV values were associated and how (increasing / decreasing)?  Thank you for your comment. This paragraph has been rewritten to improve the accuracy of our statements. We explored if NOPA is associated with different Stress-relates variables measures through HRV, but HRV is not a studied variable.

Introduction need some improvements. The aim of your research should be highlighted more in detail. Also, study motivation and a clear literature gap are not properly reported.  We appreciate all your comments. All your contributions have been taken into account and corrections have been made accordingly.

You stated: “These changes result in a decreased HRV during the presence of the stressor”. This statement is only partially correct. In fact, some of HRV parameters decrease as a response of stressors, while some other parameters (measuring sympathetic activity) can increase due to stress. Be clear reporting how HRV parameters change as a response of occupational and environmental stress sources. About this aspect, Authors can also refer to these articles. Thank you very much for your comments. This paragraph has been rewritten accordingly. The following statement has been added: “. Thus, high HRV levels is a marker of less stress. Accordingly, HRV represents the ability of the heart to respond to different contextual stimuli [14, 15]. Nonetheless, occupational stimuli such as high levels of noise or vibrations might alter the sympathetic and parasympathetic responses in airport staff or students [16, 17]”. (line 55-57).

Jalilian H, Zamanian Z, Gorjizadeh O, Riaei S, Monazzam MR, Abdoli-Eramaki M. Autonomic nervous system responses to whole-body vibration and mental workload: A pilot study. Int J Occup Environ Med 2019. https://doi.org/10.15171/ijoem.2019.1688.

Lecca LI, Marcias G, Uras M, Meloni F, Mucci N, Larese Filon F, Massacci G, Buonanno G, Cocco P, Campagna M. Response of the Cardiac Autonomic Control to Exposure to Nanoparticles and Noise: A Cross-Sectional Study of Airport Ground Staff. Int J Environ Res Public Health. 2021 Mar 3;18(5):2507. doi: 10.3390/ijerph18052507. PMID: 33802520; PMCID: PMC7967637.

You stated: “Using HRV to monitor and 60 mitigate burnout syndrome in different populations […]” It is not clear how HRV monitoring can decrease levels of stress. Explain what is the mechanism involved and working scenarios where HRV monitoring has improved workers well-being reducing stress. How can a monitoring improve well-being? This assertion is very questionable Thank you so much, the follow explanation has been added to the article: The identification of stress levels at the workplace and the application of HRV biofeedback exercises (i.e., breathing with a HRV biofeedback device) seem to be an effective way to reduce stress-related diseases and promoting health [13,19]. Further-more, HRV has been used to monitor and mitigate burnout syndrome in different populations such as white-collar workers, police officers, nurses, managers, or health professionals [20]”.

Line 71: insert citation done

The link between PA, stress levels and HRV changes is not explain at all. The theoretical background that drive your study is very weak and badly presented. Also, the hypothesis is too general and not linked with a strong theoretical framework. I suggest to carefully revise the introduction, providing the needed elements to justify your study. The effect of PA on stress and HRV must be properly presented. It is well known that PA affect autonomic response per se. So, how can attribute changes of HRV to stress and not to PA? Thanks for your recommendations, the introduction has been rewritten (lines 69-78). The relation between PA, HRV and stress has been explained.

Objective 3 is not properly described. What do you want to evaluate and how? Considering all the comments proposed by the different reviewers, we have decided to unify the second and third objectives. “to apply a questionnaire to workers measuring burnout syndrome and job satisfaction, and to compare the results with physiological stress and recovery measured objectively through Heart Rate Variability (HRV).”

Methods: How did you select the variables of interest? Explain very clearly how did you choose HRV parameters. HRV parameters are given by the BodyGuard2 device. The Firstbeat Firstbeat Technologies Ltd. software associated with BodyGuard records the data and provides the values in the variables presented in Table 1. The following state has been added: “These are the variables provided by the software associated with the device BodyGuard 2” (line 135-136).

I noticed that only RMSSD is a validated HRV measure. The other variables such as % stress, % recovery and so on are not validated HRV variables. You must provide the validation of those variables, if any.

Results: The reference you used to compare stress levels whit standard is not a scientific reference (“Firstbeat Technologies Ltd., 153 Jyväskylä, Finland”). You must provide a scientific validated reference, if any, and not a manual of use of the instrument. Is there any standard for this measure? I strongly doubt on the validity of these measures.

I attach this two references in order to answer the two previous questions at the same time. % stress, % recovery variables have been validated by independent authors. Moreover, the company has available a white paper about the validity and reliability of these technology.

  • https://assets.firstbeat.com/firstbeat/uploads/2015/10/Stress-and-recovery_white-paper_20145.pdf. Stress and Recovery Analysis Method Based on 24-hour Heart Rate Variability 2014 © Firstbeat Technologies Ltd. Published: 16/09/2014, updated: 04/11/2014
  • Also, in the follow study it is validated the use of Bodyguard for RMSSD, %stress, and % recovery. Föhr, T., Pietilä, J., Helander, E., Myllymäki, T., Lindholm, H., Rusko, H., & Kujala, U. M. (2016). Physical activity, body mass index and heart rate variability-based stress and recovery in 16 275 Finnish employees: a cross-sectional study. BMC public health16, 701. https://doi.org/10.1186/s12889-016-3391-4

Why did you include sport centres as covariates? Do you hypothesised that HRV changes on function of working centre? No, pero hay ciertas condiciones laborales que no son las mismas, como características socio-económica de los sujetos, comodidades del gimnasio, etc. We included sports centres as covariates because the fitness trainers belonged to two different centres. Our hypothesis was not that sports centres modify HRV, but each centre has different conditions: socio-economic status of the clients, fitness centre amenities, etc. And they have to be considered to analyse the specific effect on variables.

Results

I suggest to revise the results of the multiple linear regression. First of all, all the p values reported in the text as <0.05 are not the same of those reported in table 5. We greatly appreciate this reviewer's comment. Due to an error in the layout, they included the mean and p values, but in reality, these are not the reported values. As the table title indicates, the regression results are displayed as coefficients (stadards errors).

Table 5: you must report standardized beta and p values. The mean values are not required. Continuing with the previous comment, neither the mean nor the value of p is shown. The values necessary to interpret the results of the relationships (ceteris paribus) are shown: Coefficient and in parentheses, the standard errors.

For this study, it is not necessary to include the standardized coefficients, but rather the non-standardized coefficients. Standardized coefficients are useful when the objective is to know the weight of the variables in a predictive model. The objective of the unstandardizez coefficients, like the objective of the study, is to quantify the relationship between the variables in the presence of the rest of the explanatory factors. Each coefficient shows how much the dependent variable increases with an increase of one unit of the independent variable (or explanatory factor).

 Discussion

Overall, the results are poorly discussed. I suggest to follow STROBE guidelines for cross sectional studies, with those elements: Summarize key results with reference to study objectives; Discuss limitations of the study, taking into account sources of potential bias or imprecision; Give a cautious overall interpretation of results considering objectives, limitations, multiplicity of analyses, results from similar studies, and other relevant evidence; Discuss the generalizability (external validity) of the study results. Thanks for your suggestion. The discussion has been rewritten accordingly to STROBE guidelines for cross sectional studies.

I suggest to revise the interpretation of RMSSD changes. RMSSD mainly reflect parasympathetic activity, so an increase of RMSSD does not reflect an increase of stress. (see also Task Force of the European Society of Cardiology and the North American Society of Pacing and Electrophysiology. Heart Rate Variability: Standards of Measurement, Physiological Interpretation, and Clinical Use. Circulation 1996. https://doi.org/10.1161/01.CIR.93.5.1043; Shaffer F, Ginsberg JP. An Overview of Heart Rate Variability Metrics and Norms. Front Public Heal 2017; 5:258. https://doi.org/10.3389/fpubh.2017.00258.) Thank you so much for your recommendations, the description of the RMSSD function has been change in the article.

Overall, the results are not fully discussed, with a proper comparison with the field literature. A discussion of the generalizability should be highly appreciated in this section. You should also carefully explain what is the specific contribution that your findings bring to literature and knowledge in this area. Please be very clear on what your study adds, exactly how it extends previous knowledge. Thanks for your recommendations. The text overall has been rewritten.

Conclusions are only partially supported by your results. In details, HRV is affected by a lot of occupational and environmental factors, so a weak correlation between self-reported stress and HRV is not surprising. Increasing work demand to manage stress is against a lot of previous knowledge that clearly assert that increasing job demand strongly affect perceived stress. Sorry, this is a translation error. When we talk about labor demands, we are referring to working conditions. In general, the working conditions of fitness instructors are very poor (irregular schedules, poor working hours, low wages, class types, etc). We suggest that a change in the working conditions of this population could lead to greater job satisfaction. This expression has been changed throught the document.

Round 2

Reviewer 1 Report

Dear authors,

I have read the revised version and it has gained a lot.

I have only two points:

  1. please do not talk about a questionnaire on "job satisfaction" the questions are on "working conditions" or maybe on "working conditions and satisfaction".
  2. in the abstracts last sentence should start with "better" not "higher"

good luck,

Mathias Nübling

Author Response

Dear reviewer, 

Thank you very much for your contributions, these have been corrected and can be seen in red, in the main document. 

Again, I am very grateful to you, 

Marín 

Reviewer 2 Report

Dear Authors

The paper has been substantially improved with respect to the first version.

All my concerns have been properly addressed.

Best Regards

Author Response

Dear reviewer, 

Thank you very much for your contributions, 

I am very grateful to you, 

Marín